# Subirrigation Effects on Larch Seedling Growth, Root Morphology, and Media Chemistry

**Fangfang Wan [1], Amy L. Ross-Davis [2], Wenhui Shi [3], Christopher Weston [4], Xiehai Song [1], Xiaochao Chang [1], Anthony S. Davis [5], Yong Liu [1,\*] and Fei Teng [1]**

1   Key Laboratory of Silviculture and Conservation, Ministry of Education, Beijing Forestry University, Beijing 100083, China; wanfangfang1991@163.com (F.W.); sxiehai@163.com (X.S.); changxiaochao@bjfu.edu.cn (X.C.); tengf7@chinaunicom.cn (F.T.)
2   Forest Ecosystems and Society; College of Forestry; Oregon State University; 109 Richardson Hall, Corvallis, OR 97331, USA; rossdava@oregonstate.edu
3   State Key Laboratory of Subtropical Silviculture, College of Forestry and Biotechnology, Zhejiang A&F University, Hangzhou 311300, China; shiwenhui2008@163.com
4   School of Ecosystem and Forest Science, the University of Melbourne, Creswick 3363, Victoria 3010, Australia; weston@unimelb.edu.au
5   Forest Engineering, Resources, and Management; College of Forestry; Oregon State University; 109 Richardson Hall, Corvallis, OR 97331, USA; anthony.davis@oregonstate.edu
\*   Correspondence: lyong@bjfu.edu.cn; Tel.: +86-010-62338994

**Abstract:** Subirrigation (SI), where water is provided to container seedlings from below and rises through the growing media via capillary action, is regarded as an environmentally-responsible method of delivering water and fertilizer to nursery-grown plants, resulting in more uniform crops and improved production efficiency. While a concern around adopting this method is that a potential higher salt concentration in the upper layers of growing media under SI may inhibit root growth and result in decreased plant quality, few studies have focused on how root morphology is altered by SI. Therefore, a balanced two-factor factorial design with three rates of fertilization (50, 100, and 150 mg N seedling$^{-1}$) and two irrigation methods (SI or overhead irrigation (OI)) was used to examine the growth response of Prince Rupprecht's larch (*Larix principis-rupprechtii* Mayr) seedlings for one nursery season. Associated changes between rhizosphere electrical conductivity (EC) and root morphology of different root size classes were analyzed. Results show that (1) height, root-collar diameter, and root volume were similar between seedlings grown under SI and OI. However, (2) compared to seedlings receiving OI, SI-seedlings had less root mass, length, and surface area but greater average root diameter (ARD). (3) Morphological differences were evident primarily in root diameter size classes I–III (D $\leq$ 1.0 mm). (4) Fertilizer rate influenced root length and surface area up to 130 days after sowing but affected ARD throughout the growing season such that seedlings treated with 50 mg N had smaller ARD than seedlings treated with 100 mg N. (5) As the growing season progressed, SI-media had significantly higher EC compared to OI-media and EC increased with increasing fertilizer rate under SI but not under OI. These results indicate that SI can produce larch seedlings of similar height and root collar diameter (RCD) compared to OI, but root systems are smaller overall with fewer small-diameter roots, which may be related to high EC levels in SI-media, which is exacerbated by the use of high rates of fertilizer. Therefore, the EC in the media should be monitored and adjusted by reducing fertilizer rates under SI.

**Keywords:** water use efficiency; seedling quality; runoff conservation; forest nursery production

---

## 1. Introduction

High demand for container-grown tree seedlings for forestry and conservation uses has an often-untracked resource need [1]: A substantial demand for irrigation water during nursery production. Conventional overhead irrigation (OI) has traditionally been the most widely used method to produce tree seedlings, as it is simple and relatively low-cost [2–4]. However, OI is inefficient in that water is not evenly distributed across the crop and considerable run-off may contaminate fresh water resources [5,6]. As such, more sustainable irrigation methods that maximize water and fertilizer use efficiency are being sought.

Subirrigation (SI), a closed-cycle system where water is provided to container seedlings from below and rises through the growing media via capillary action, may be regarded as one of the most promising methods of irrigation in forest tree seedling nurseries. The application of this technology has been examined for a variety of horticultural, agricultural, and forestry crops, e.g., [2,7–16], and results indicate that there is the ability to produce seedlings of comparable quality to those grown using OI [7,8,10]. These studies have examined crop production under SI across a variety of container sizes, water and fertilization regimes, substrates, and irrigation durations and frequencies. Most studies examined water and fertilizer consumption, media chemistry and physical properties, and seedling morphology (e.g., height, diameter, biomass and leaf area), nutrition, and mortality to evaluate the efficacy of SI. From this body of research, SI has many apparent benefits such as improved water and fertilizer use efficiency [2,17], decreased liverwort and moss growth [6], reduced labor needs [18], and reduced foliar disease [19].

Despite such promise, demonstrated and theoretical concerns exist that limit the broader application of SI. For example, SI may promote root disease if water is recycled without disinfection [19,20], however there is evidence that partial saturation in a subirrigated regime may reduce disease incidence compared to standard SI saturation protocol [21]. Due to the nature of water delivery (capillary action from below) and loss (transpiration and evaporation), it is reasonable to expect that SI will alter the rhizosphere's chemical properties compared to OI.

Salt concentration, measured as electrical conductivity (EC), and particularly in the upper portion of the growing media, has been found to increase in SI systems [2,11,22–26], although there is evidence to suggest that elevated EC in the upper media layer can be lowered immediately by an overhead freshwater flush [10]. The problem may, however, be exacerbated by the use of high rates of fertilizer, leading to decreased seedling root mass and greater mortality [25,26].

Jacobs and Timmer described how fertilization can dramatically alter the rhizosphere EC level and influence root system growth and function, thereby impacting the ability of plants to absorb water and nutrients from soil [27]. Thus, soil EC must be monitored regularly to evaluate fertilization protocols and more attention must be given to underground seedling growth to assess plant health. Previous studies have reported on total root mass, showing similar or greater total root mass in SI plants compared to those in OI, however, stratified sampling from different media layers to observe the influence of EC values on root mass showed root mass of the upper media layer decreased with higher EC values, while total root mass was unchanged [25,26]. Thus, total root mass alone may not be adequate to assess root system growth and may not capture the heterogeneity in growing media.

Fine roots typically influence water and nutrient uptake, while coarse roots determine the capacity of water and nutrient transport from fine roots to aboveground tissues [28,29]. Many studies have reported that root morphology (e.g., diameter, length, surface area, specific root length, and specific root surface area) reflects the soil environment including N availability, water, and soil temperature; and is an important indicator of most important physiological activities like respiration and transpiration [29–33]. For example, root diameter is closely related to root respiration, which drives root growth [34–36]. Specific root length (SRL), the length per weight of a root segment, is regarded as one of the most useful indicators of environmental change such as fertilization, physical disturbance of soil, and varying moisture and temperature levels [37].

While previous research has demonstrated changes to the chemical and physical properties of growing media under SI, very few experiments have directly studied changes to root morphology [38,39]. Prince Rupprecht's larch (*Larix principis-rupprechtii* Mayr) is one of the main reforestation species of north China with the advantages of being fast-growing, of good timber quality, widely used, and resistant to decay, making it a strong candidate for study in a SI system. This study focused on larch root morphology and EC changes throughout the growing media to examine the effect of SI with controlled-release fertilizer (CRF) addition compared to conventional overhead irrigation of seedlings. We hypothesized that: (1) SI would produce similarly sized seedlings compared to OI; (2) SI-seedlings would have significantly different root morphology relative to OI-seedlings; (3) SI would require less fertilizer than OI to produce seedlings of similar morphological attributes.

## 2. Materials and Methods

### 2.1. Plant Material and Nursery Cultural Conditions

Larch seeds were collected in November 2014 from Longtou mountain seed stand in Weichang country, Chengde city of Hebei province (42°05′ N, 117°24′ E), placed in a cotton fabric bag, and stored at 2 °C until the experiment began the following March. Seeds were germinated on a moist sand bed covered with gauze until about 30% of the seeds broke seed coat. Then, all the seeds were sown (2 seeds in each container) into 164 cm$^3$ Ray Leach "Cone-tainer"™ containers (3.8 cm top diameter × 21 cm length; Stuewe & Sons, Inc., Tangent, Oregon, USA) and grown in a greenhouse on the Miaofeng Mountain Experimental Base of Beijing Forestry University (39°54′ N, 116°28′ E). Containers were assembled into 24 trays (each holding 49 containers) and filled with a mix of 70% peat (Pindstrup Seeding, Ryomgaard, Denmark; pH = 6.0; Screening, 0–6 mm) and 30% perlite (5 mm diameter; Xinyang Jinhualan Mining Co., Henan, China) on 15 April 2015. Osmocote NO.5 (14N-5.7P-10.8K; Scotts®, Marysville, OH, USA) 5–6 month controlled-release fertilizer (CRF) was incorporated into the growing media at one of three rates: 50, 100, 150 mg nitrogen (N) per seedling (equal to 0.357, 0.714, 1.071 g CRF per seedling, respectively); which met low, medium and high application rates of soluble nitrogen (N) fertilizer for container-grown larch [38]. All containers were watered by hand using a hose with nozzle from overhead until seedlings had grown two rounds of needles, which occurred within four weeks of sowing. Seedlings were then thinned to one per cell.

A total of 24 trays were arranged in a balanced two-factor factorial design with 196 seedlings of each treatment combination (totaling 1176 seedlings) from three fertilizer rates × two irrigation methods. Seedlings of each fertilizer rate treatment were then randomly divided into two groups: (1) hand-watered from overhead and (2) watered from below, known as subirrigation (SI). Seedlings were grown for 22 weeks under mean day/night temperatures of 28/16 °C and ambient light conditions. After 22 weeks, plants were moved outdoors to initiate hardening.

SI was accomplished by placing each tray of seedlings into a plastic box (Citylong, 54 cm long, 40 cm wide and 31 cm deep; Beijing Citylong plastic products co. LTD, Beijing, China) filled to a depth of approximately 15 cm with water for 20 minutes (the amount of time required to return to saturation) at a frequency determined by the gravimetric water content (GWC), according to the growth phase. Boxes were marked by fertilizer application rate to avoid the confusion of different treatments. The irrigation schedule was determined by GWC and the target weight was calculated using the "Scientist Technique" [40]. Seedlings were irrigated when GWC reached 85% of field capacity during the establishment phase (weeks 1–4). During the rapid growth phase (weeks 5–15), irrigation was scheduled at a GWC of 75%. Finally, irrigation was scheduled at a GWC of 65% when the seedlings were in the hardening phase (weeks 16–22). When subirrigated to field capacity, trays were put on the surface of the containers and secured by two sticks to allow the extra water leaching from the media to flow back into the water boxes. Water in the boxes was retained until the final watering. Boxes were covered with black shade screen to protect the water from moss and algae growth.

### 2.2. Root Morphology and EC Measurement

Twenty seedlings from each fertilizer rate-irrigation system treatment were randomly chosen to measure height and root-collar diameter (RCD) at 70, 102, 130, 160, 183, and 203 days after sowing (DAS; 23 June, 26 July, 23 August, 28 September, 21 October, 10 November). Six seedlings were harvested on each date and separated into shoots and roots. The subsamples of roots were gently washed to remove growing media and an image of the root system was acquired by scanning with the Epson Twain Pro high-quality scanner at 400 dpi (Epson Co., Ltd., Beijing, China). The root system was divided into five classes according to root diameter [38,39]: 0–0.2 mm (class I), 0.2–0.5 mm (class II), 0.5–1.0 mm (class III), 1.0–2.0 mm (class IV), and >2.0 mm (class V). Scanned images of the roots were used to determine total root length, root surface area, root volume, and average root diameter (ARD) using WinRHIZO software (Régent Instruments Inc., Quebec, Canada). After scanning, subsamples were dried at 65 °C to constant mass (approximately 72 h) and weighed. Specific root length (SRL) was calculated for each root system as the quotient of total root length (cm) and root dry mass (g).

To capture EC trends in container media, the midpoint of the sensors of Fieldscout EC probe were inserted at three depths: top (3 cm), middle (10 cm), and bottom (17 cm) as measured from the top of the container at 75, 100, 127, 157 and DAS (27 June, 24 July, 21 August, 20 September). Fifteen seedlings were randomly selected from each treatment one hour after irrigating to field capacity to ensure accurate data collection [8,41].

### 2.3. Statistical Analysis

Two-factor analysis of variance (ANOVA) was used to analyze the effects of irrigation method and fertilizer rate and their interactions using the SPSS 18.0 (SPSS®, Chicago, IL, USA). Pairwise comparisons within main effects were analyzed using the Tukey HSD method ($p < 0.05$). When assumptions for equal variances and normality were not met, data were log-transformed to meet the required assumptions for analysis. Graphs were produced using SigmaPlot 12.5 (Systat Software, Inc., San Jose, CA, USA).

## 3. Results

### 3.1. Effects of Irrigation and Fertilizer Rate on Larch Seedling Morphology

At the end of the growing season (203 DAS), there were no significant irrigation × fertilizer rate effects on seedling morphology but irrigation and fertilizer rate had significant independent effects (Table 1). Root mass, root length, root surface area, average root diameter (ARD), and specific root length (SRL) differed significantly between SI and OI seedlings (Table 1). OI seedlings had 13% greater root mass, 66% greater root length, 37% greater root surface area, and 40% greater SRL compared to SI seedlings; SI seedlings had 23% greater ARD than OI seedlings (Table 1). Irrigation method had no significant effect on seedling height, root collar diameter (RCD), stem mass, whole seedling mass, or root volume (RV) by the end of the growing season (203 DAS).

At 203 DAS, seedlings treated with 100 mg N had 19% greater ARD than those treated with 50 mg N (Table 1), but beyond that, fertilizer rate had no significant effect on final plant morphology. Fertilizer rate had significant effects on several measures of larch seedling morphology early in the growing season (up to 160 DAS), however those dissipated over time (Table S1 and Figure 1). And the interaction of irrigation and fertilizer rate was only observed on root volume, ARD and SRL at 160 DAS (Table S1 and Figure S1).

**Table 1.** Larch seedling morphology at time of final harvest (203 days after sowing; 10 November). Column means ± SE followed by different letters within a given treatment differ significantly according to Tukey's honestly significant difference test at α = 0.05. *P*-values for main effects of irrigation and fertilizer rate and their interaction on larch seedling morphology at time of final harvest.

| Source | Growth Parameter | | | | | | | | | |
|---|---|---|---|---|---|---|---|---|---|---|
| | Height (cm) | Root Collar Diameter (mm) | Stem Mass (g) | Root Mass (g) | Whole Seedling Mass (g) (Stem + Root) | Root Length (cm) | Root Surface Area (cm$^2$) | Root Volume (cm$^3$) | Average Diameter (mm) | Specific Root Length (cm/g) |
| Irrigation (I) | | | | | | | | | | |
| Overhead irrigation (OI) | 28.94 ± 0.36a | 3.13 ± 0.04a | 0.58 ± 0.02a | 0.61 ± 0.01a | 1.19 ± 0.03a | 869.96 ± 72.83a | 152.4 ± 18.63a | 2.27 ± 0.35a | 0.57 ± 0.02b | 1368.67 ± 136.82a |
| Subirrigation (SI) | 29.34 ± 0.29a | 3.20 ± 0.06a | 0.63 ± 0.02a | 0.54 ± 0.02b | 1.17 ± 0.04a | 524.87 ± 59.22b | 111.46 ± 8.92b | 1.92 ± 0.12a | 0.70 ± 0.03a | 981.07 ± 111.49b |
| Fertilizer Rate (F) | | | | | | | | | | |
| 50 mg·N·seedling$^{-1}$ | 29.36 ± 0.23a | 3.12 ± 0.07a | 0.62 ± 0.03a | 0.58 ± 0.02a | 1.20 ± 0.05a | 736.88 ± 64.82a | 130.02 ± 11.67a | 1.88 ± 0.21a | 0.57 ± 0.04b | 1269.85 ± 104.20a |
| 100 mg·N·seedling$^{-1}$ | 29.04 ± 0.42a | 3.16 ± 0.07a | 0.60 ± 0.02a | 0.55 ± 0.02a | 1.15 ± 0.04a | 735.90 ± 133.06a | 151.39 ± 24.64a | 2.51 ± 0.37a | 0.68 ± 0.04a | 1320.78 ± 223.47a |
| 150 mg·N·seedling$^{-1}$ | 29.02 ± 0.53a | 3.20 ± 0.05a | 0.59 ± 0.03a | 0.59 ± 0.02a | 1.18 ± 0.03a | 558.69 ± 87.05a | 114.37 ± 18.53a | 1.89 ± 0.33a | 0.65 ± 0.03ab | 933.99 ± 134.86a |
| *p*-values | | | | | | | | | | |
| Irrigation (I) | 0.434 | 0.363 | 0.122 | 0.004 | 0.695 | 0.003 | 0.023 | 0.323 | 0.001 | 0.028 |
| Fertilizer Rate (F) | 0.822 | 0.692 | 0.723 | 0.370 | 0.695 | 0.448 | 0.214 | 0.262 | 0.025 | 0.125 |
| I × F | 0.950 | 0.576 | 0.754 | 0.986 | 0.869 | 0.100 | 0.075 | 0.215 | 0.703 | 0.132 |

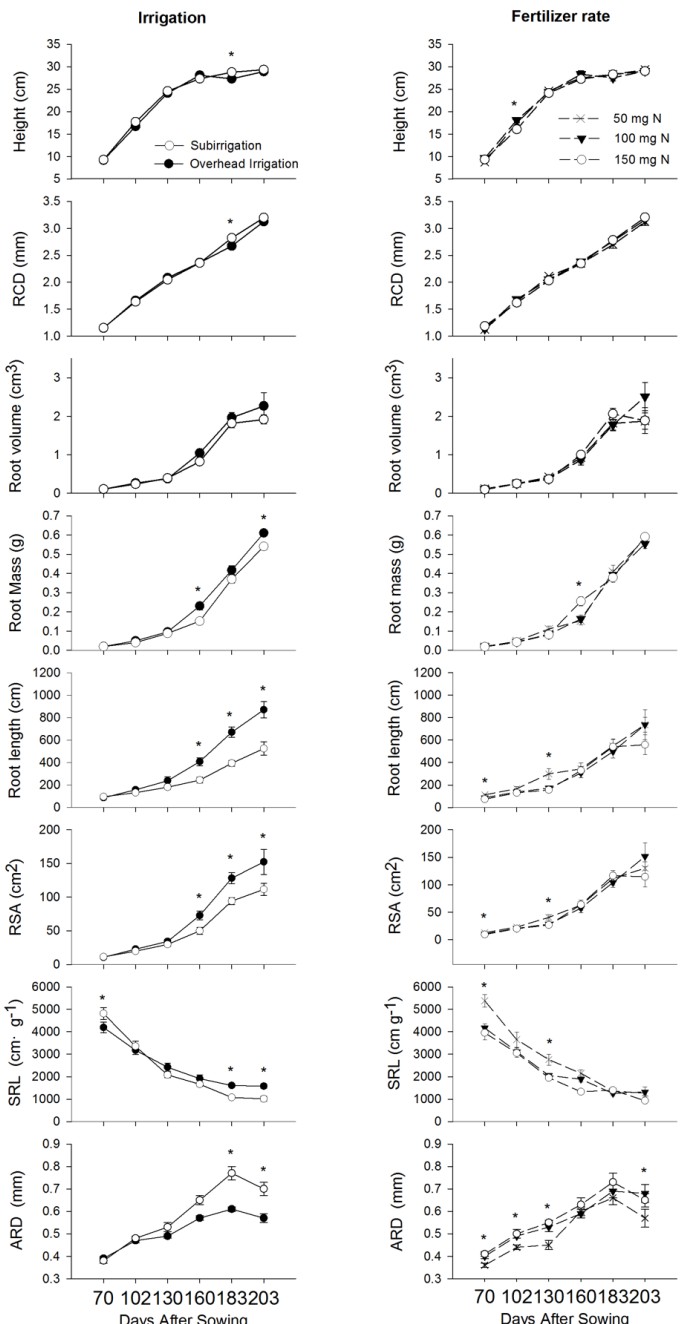

**Figure 1.** Height, RCD (root collar diameter), root volume, root mass, root length, RSA (root surface area), SRL (specific root length) and ARD (average root diameter) for 70 to 203 days-old larch seedlings watered either from overhead irrigation or subirrigation (left panel) or grown with three fertilizer rates (right panel). An asterisk indicates a significant difference with a given irrigation treatment according to Tukey's honestly significant difference test at α = 0.05.

### 3.2. Effects of Irrigation and Fertilizer Rate on Growing Media EC

At each assessment during the growing season (i.e., 75, 100, 127, and 157 DAS), the main effects of fertilizer rate and irrigation method interacted significantly to affect growing media EC at all three media depths measured (Table 2 and Figures 2 and 3). As the growing season progressed, SI media had significantly higher EC compared to OI media across all fertilizer rates at all three layers investigated (Figure 3). This difference was particularly pronounced at the 100 mg N and 150 mg N fertilizer rates in the middle and top media layers (Figure 3). Initially, EC increased with increasing fertilizer rates in the

middle and bottom media layers under OI. However, there was no significant difference in OI-media EC across fertilizer rates as the growing season progressed (Figure 2). Under SI, EC generally increased with increasing fertilizer rates in all three media layers throughout the entire growing season.

**Table 2.** *p*-values for main effect of fertilizer rate and irrigation and their interaction on media EC across the growing season.

| Time | Source | *p* Value | | |
|---|---|---|---|---|
| | | Top | Middle | Bottom |
| 75 DAS | Irrigation (I) | <0.001 | 0.001 | <0.001 |
| | Fertilizer Rate (F) | <0.001 | <0.001 | <0.001 |
| | I × F | <0.001 | 0.002 | <0.001 |
| 100 DAS | Irrigation (I) | <0.001 | <0.001 | <0.001 |
| | Fertilizer Rate (F) | <0.001 | <0.001 | <0.001 |
| | I × F | <0.001 | <0.001 | <0.001 |
| 127 DAS | Irrigation (I) | <0.001 | <0.001 | <0.001 |
| | Fertilizer Rate (F) | <0.001 | <0.001 | <0.001 |
| | I × F | <0.001 | <0.001 | 0.002 |
| 157 DAS | Irrigation (I) | <0.001 | <0.001 | <0.001 |
| | Fertilizer Rate (F) | <0.001 | <0.001 | <0.001 |
| | I × F | <0.001 | <0.001 | <0.001 |

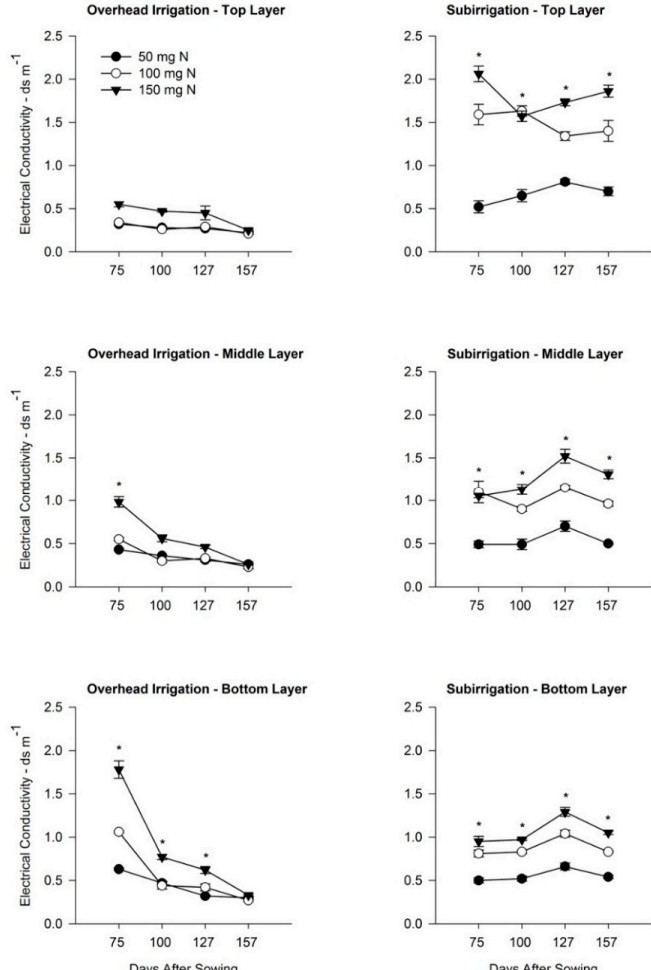

**Figure 2.** Electrical conductivity (EC, ds m$^{-1}$) from days 75 to 157 of growing media as measured at three depths (top (3 cm), middle (10 cm), and bottom (17 cm)) in the seedling containers for each combined irrigation method and fertilizer rate treatment. An asterisk indicates a significant difference within a given irrigation treatment and media layer according to Tukey's honestly significant difference test at α = 0.05.

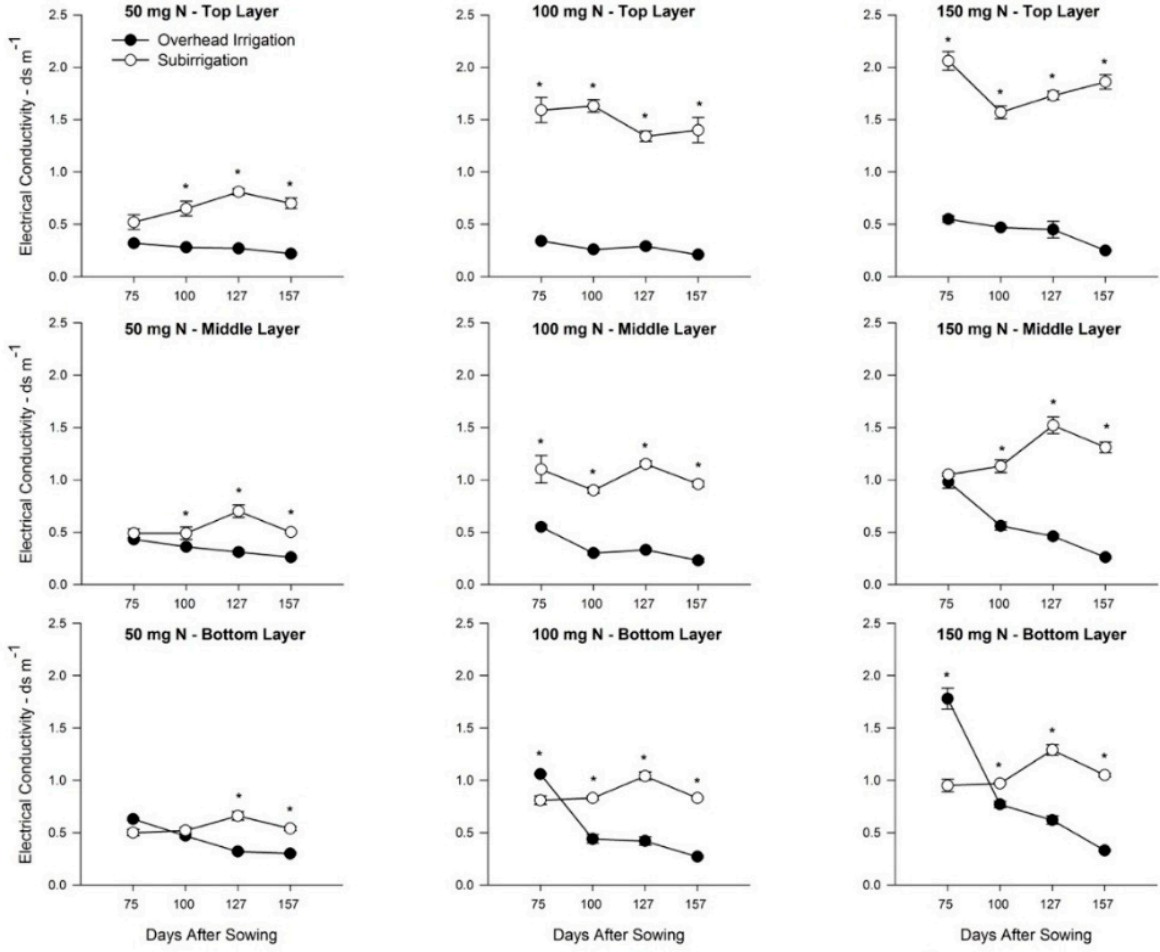

**Figure 3.** Electrical conductivity (EC, ds m$^{-1}$) from days 75 to 157 of growing media as measured at three depths (top (3 cm), middle (10 cm), and bottom (17 cm)) in the seedling containers for each combined irrigation method and fertilizer rate treatment. An asterisk indicates a significant difference within a given fertilizer rate treatment and media layer according to Tukey's honestly significant difference test at $\alpha = 0.05$.

### 3.3. Effects of Irrigation and Fertilizer Rate among Root Diameter Classes

At final harvest (203 DAS), roots in diameter classes I–III accounted for ~90% of total root length, ~70% of total root surface area, and ~30% of the total root volume. For roots in diameter classes I and II, length, surface area, and volume were significantly lower under SI than under OI (Table 3 and Figure 4). Further, roots in diameter class I had reduced volume and those in class II had reduced length, surface area, and volume when treated with 150 mg N compared to those treated with 50 mg N (Table 3 and Figure S2). For roots in diameter class III, a significant irrigation × fertilizer rate effect was detected for root length, surface area, and volume (Table 3). While there was no difference in length, surface area, or volume of class III roots across fertilizer rates for SI seedlings, length, surface area, and volume were significantly reduced among OI seedlings when treated with 50 mg N compared to those treated with 100 mg (Figure S3). Furthermore, length, surface area, and volume of class III roots were significantly lower among SI seedlings compared to OI seedlings only when treated with 100 mg N (Figure S3). There was no effect of irrigation or fertilizer rate on roots with diameters > 1.0 mm (Table 3 and Figure 4).

**Table 3.** *p*-values for main effect of fertilizer rate and irrigation and their interaction on root length surface area and volume of different class.

| Parameter | Source | Class I | Class II | Class III | Class IV | Class V |
|---|---|---|---|---|---|---|
| Root length | Irrigation (I) | 0.004 | <0.001 | 0.023 | 0.893 | 0.939 |
| | Fertilizer Rate (F) | 0.102 | 0.013 | 0.025 | 0.297 | 0.441 |
| | I × F | 0.461 | 0.229 | 0.001 | 0.435 | 0.738 |
| Root surface area | Irrigation (I) | 0.003 | <0.001 | 0.023 | 0.925 | 0.926 |
| | Fertilizer Rate (F) | 0.062 | 0.024 | 0.061 | 0.323 | 0.485 |
| | I × F | 0.593 | 0.214 | 0.004 | 0.494 | 0.759 |
| Root volume | Irrigation (I) | 0.003 | <0.001 | 0.040 | 0.956 | 0.334 |
| | Fertilizer Rate (F) | 0.047 | 0.030 | 0.061 | 0.350 | 0.285 |
| | I × F | 0.684 | 0.136 | 0.005 | 0.560 | 0.232 |

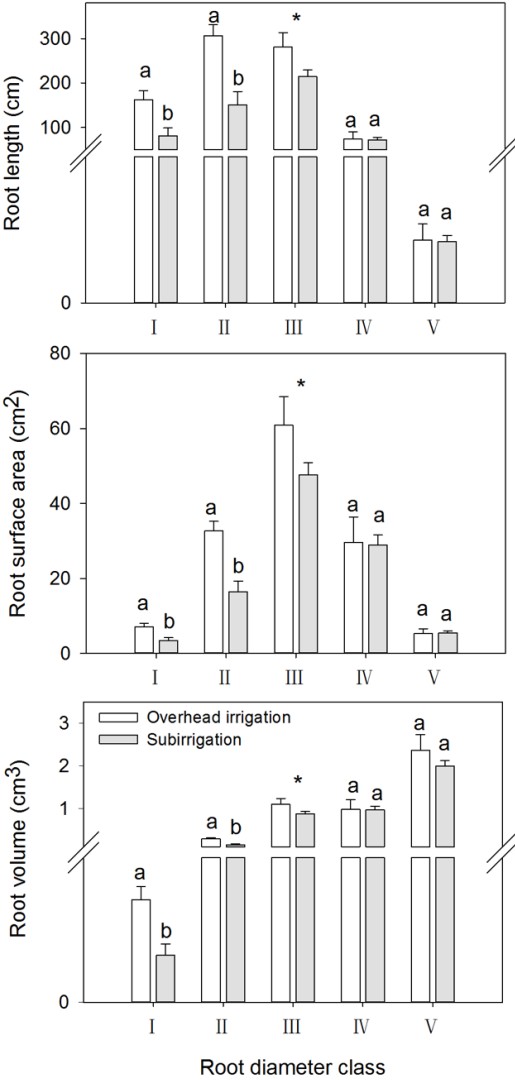

**Figure 4.** Larch seedling root length, root surface area, and root volume among root diameter classes at final harvest (203 days after sowing; 10 November). Treatments marked with different lower-case letters differ statistically at each root diameter class according to Tukey's honestly significant difference α = 0.05. An asterisk means a significant interaction for class III roots and see Figure S3.

## 4. Discussion

### 4.1. Effects of Irrigation on Shoot Morphology and Root Architecture

Although seedling shoot morphology was similar between irrigation methods, SI seedlings developed significantly different root architectures, with shorter total and specific lengths, less surface area, and greater average diameters compared to OI seedlings. These differences became apparent well into the growing season (from 160 DAS on) and were particularly pronounced among small diameter roots (≤1.0 mm). These findings are largely consistent with previous studies. For instance, Dumroese et al. and Davis et al. reported no significant effect of irrigation method on seedling RCD or height for 'ohi'a (*Metrosideros polymorpha* Gaud.), northern red oak *(Quercus rubra* L.), or trembling aspen (*Populus tremuloides* Michx.) [6,8,10,11]; and Davis et al. also reported a similar RV between SI and OI trembling aspen seedlings [11]. Effects of SI on root mass are variable, e.g., [2,26,42] and likely dependent on species, fertilizer rate, and length of the growing season. Specific measures of root morphology, particularly of small diameter roots, appear to provide a more sensitive indicator of rhizosphere conditions than root mass alone, as suggested by Hirano et al. [43].

### 4.2. Effects of Fertilizer Rates on Roots Morphology

Fertilizer rate had pronounced effects on root morphology among small diameter classes (≤1.0 mm). Doi et al. and Wang et al. also reported that N addition influenced growth of fine roots but had no effect on coarse roots for *Chamaecyparis obtusa* and *Pinus tabuliformis*, respectively [28,29]. These phenomena indicate that fine roots are more sensitive to changes to the rhizosphere and thus are essential parameters to evaluate water and nutrient uptake [44,45]. Further, fertilizer rate significantly influenced root morphology primarily early in the growing season (up to 130 DAS), with reduced root length and root surface area at the highest CRF rate (150 mg N seedling$^{-1}$), consistent with other reports that show decreased root growth under higher fertilizer rates [29,46,47]. For example, Wang et al. reported a significant decrease of fine root number, length, and biomass of *P. tabuliformis* with N addition (0–12 g N m$^{-2}$ y$^{-1}$) [29]. Nonetheless, many studies also report that fertilizer could promote root growth in terms of root length and root surface area [48–50]. Shi et al. found that a high fertilizer rate (100 mg N seedling$^{-1}$) increased both surface area and first order lateral roots of one year-old *Quercus variabilis* seedlings, compared with no fertilizer [50]. These conflicting results may be due to species-specific responses and/or to different fertilizer rates among studies.

The only persistent effect of fertilizer rate was on ARD. At final harvest (203 DAS), seedlings receiving 50 mg N had smaller ARD than seedlings receiving 100 mg N. Greater root length, surface area and root volume are not always indicative of reduced ARD, as observed in previous studies [28,31,51,52]. Looking specifically within individual root diameter classes, fertilizer rate had a significant effect on fine roots (classes I, II, and III) but no effect on coarse roots (>1 mm diameter). Specifically, seedlings receiving 50 mg N had greater root length (class II), surface area (class II), and volume (class I and II) compared to those receiving 150 mg N. However, class III roots had reduced length, surface area, and volume among seedlings treated with 50 mg N compared to those treated with 100mg N under OI (with no fertilizer effect apparent under SI). Thus, the low fertilizer rate which favored the development of small diameter roots (≤0.5 mm diameter) meant an overall reduced ARD at final harvest.

### 4.3. Root Morphology Response to Growing Media EC Changes

We suspect that these changes in root morphology among SI seedlings may be a morphological response to rhizosphere conditions to facilitate acquisition of available nutrients to maximize whole-seedling growth. SI-media EC remained high throughout the growing season, indicative of a consistent supply of adequate nutrients. This is not surprising given the closed-nature of SI and the retention of nutrients over the growing season compared to OI; thus, seedlings did not require long roots with greater surface areas to absorb adequate nutrients. Roots of SI seedlings were significantly

shorter with larger ARD, likely leading to a greater ability for water and nutrient transport since coarse roots with shorter, larger diameters are the main organs for water transport [45,53–55]. Similarly, Wang et al. reported that increasing N rate decreased root number, length, and mass but increased root respiration rate and root hydraulic conductance of fine roots and improved water-transport by influencing coarse roots of *P. tabuliformis* seedlings [29].

Though we did not measure available N or other nutrient elements in the growing media, EC throughout the media profile was measured to characterize the buildup of fertilizer salts. EC is an effective indicator of fertilizer salts and increases exponentially with fertilizer addition [4]. In our study, higher EC in the top media layer under SI was consistent with results noted in other SI trials [6,8,10,15,26,27]. Fortunately, we did not observe significant impairment to seedling growth likely because media EC, which ranged from 0.50–2.01 ds m$^{-1}$, remained within the tolerable range for larch seedlings [38].

EC was significantly greater for SI media than for OI media with the exception of the bottom layer at 75 DAS. This notable exception is likely due to the initial release and transport of CRF to the bottom layer of the growing media under conventional OI. Pinto et al. noted that EC of OI coneflower leachate was initially high but dropped 53% and remained low for the remainder of the growing season [26], which is concordant with our hypothesis. Unlike this sharp decline in OI media EC over time, EC of SI media did not decline, likely the result of recycling water (and dissolved nutrients). The higher EC levels maintained throughout the SI media suggests the potential to decrease fertilizer quantities recommended for crop production [10]. In our current study, even SI media treated with the lowest CRF rates (50 and 100 mg N seedling$^{-1}$), had significantly higher EC values than the OI media treated with the highest CRF rate (150 mg N seedling$^{-1}$). This trend became more pronounced as the growing season progressed. Thus, there is evidence to support our hypothesis that fertilizer rate can be reduced under SI regimes compared to OI regimes.

## 5. Conclusions

High fertilizer rates and SI (likely due to higher growing media EC) negatively influenced fine root growth of seedlings. However, considering SI can improve water and nutrient efficiency, and can produce seedlings with higher uniformity in biometric parameters such as height, root diameter, stem mass, whole seedling mass and root volume without an effect on coarse roots compared to OI, SI could serve as an alternative irrigation method to yield quality container seedlings if coupled with a reasonable fertilizer regime for the species to prevent salt accumulation in the growing media. Monitoring EC across the growing season will allow growers to respond to accumulated salts by providing an overhead fresh water flush, for example, when needed. In our current study, we did not measure root physiological activities, e.g., root respiration rate or root hydraulic conductance, which may be more sensitive indicators for water and nutrient uptake and transport. We suggest that future studies examine root physiological responses to SI.

**Supplementary Materials:** The following are available online at http://www.mdpi.com/1999-4907/10/1/38/s1, Figure S1: Larch seedling total root volume, average root diameter, and specific root length at 160 days after sowing; Figure S2: Effect of fertilizer rate on volume of class I and class II roots and length and surface area of class II roots at 203 days after sowing; Figure S3: Length, root surface area, and root volume of diameter class III roots for which there was a significant interaction of fertilizer rate and irrigation method; Table S1: P-values for main effect of irrigation and fertilizer rate and their interaction on larch seedling morphology throughout the growing season.

**Author Contributions:** Conceptualization and methodology, Y.L., F.W. and W.S.; software, F.W. and A.L.R.-D.; formal analysis, F.W.; investigation, F.W., X.S., X.C. and F.T.; writing—original draft preparation, F.W.; writing—review and editing, A.L.R.-D., A.S.D., Y.L. and C.W.; funding acquisition, Y.L.

**Funding:** This study was supported by the "948" Plan of China [grant number 2012-4-66].

**Acknowledgments:** We gratefully acknowledge the reviewers for their insightful comments of the manuscript, thank other team members for help with the experiment, and thank the staff in Jiufeng for managing the nursery.

**Conflicts of Interest:** The authors declare no conflict of interest.

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
