# Peer review of "Subirrigation Effects on Larch Seedling Growth, Root Morphology, and Media Chemistry"

_forests, doi:10.3390/f10010038_

Reviewer 1 Report

I found the manuscript to be well written and containing information which should be useful to nursery managers and any who may be concerned with seedling quality.Some items for your attention:

- give the full common name = Prince Rupprecht's larch -- L. 25 and L. 93

L 43 -- "T" - ?

L. 46 - ....low cost below -- what does "below" mean in this sentence?

L. 103 -- a little more detail would be helpful. How many seeds? You state that 30%  = 1176. Did you collect an exact number? How many different trees were involved? Any possibility that seed source could influence your results?

L. 127 -- GWC is not defined until L. 129 ( unless I missed it)

L. 139 -- I do not think "sowing" is the correct term. As per L. 107 -- these were transplanted.

L. 165, L. 170, L. 255, L. 308, and L. 325- delete "larch" ( all the seedlings in this study were larch)

Author Response

Dear Editors and Reviewers:

   Thank you for your letter and for the reviewers’ comments concerning our manuscript entitled “Subirrigation Effects on Larch Seedling Growth, Root Morphology, and Media Chemistry” (ID: forests-409234). The comments are all valuable and very helpful for revising and improving our paper, as well as highlighting the significance of our research. We have studied comments carefully and have made corrections, which we hope meet with your approval. Revised portions are marked in the paper using the “Track Changes”. The main corrections in our manuscript and our response to the reviewers’ comments are indicated below. 

   We hope the revisions in the manuscript and our accompanying responses meet your approval. We will be happy to work with you to resolve any remaining issues. 

    We shall look forward to hearing from you at your earliest convenience. 

    Thank you and best regards.

    Yong Liu, Ph.D., Professor 

  Key Laboratory of Silviculture and Conservation, Ministry of Education, College of Forestry, Beijing Forestry University, Beijing 100083, P. R. China 

    Phone: +86-010-62338994

Reviewer 2 Report

The manuscript clearly and in detail shows results of an interesting research on influence of subirrigation on larch seedlings. This research facilitates nursery methods to produce more even and more quality larch seedlings. Thus, the research is very important for enhancement of production cycle. This makes the research complex, interesting and original. The research results are significant from the perspective of forest management and forest conservation, since the quality nursery stock is the prerequisite of successful silvicultural activities. The research provides original results, so the manuscript belongs to the category of original research papers. The text is comprehensive and authors have fully followed journal instructions for authors.

More detailed comments to the authors:

Title:

Title clearly describes the topic of the research.

Abstract:

Abstract is missing section on materials and methods used.

L28 Please shortly state what were methods used in the research.

L 39 Statement L321 – 322 which refers to lowering fertilization rate during SI regime is important for the use of SI and should be mentioned in the abstract.

Introduction:

L 43 “T High” should be corrected.

L 46 Delete “below”.

Materials and methods:

Written clear and comprehensive, with enough details needed for repetition of the experiment.

Results:

Authors should still reconsider the number of tables and graphs they have provided in the manuscript. Strong suggestion is to further reduce the Figures, leaving only those which directly answer the research hypotheses. Other Figures/Tables should also be placed in the Supplement. Authors can also consider some other graphical option of presenting these valuable and important results. Those could also present more clearly dynamic of measured parameters, and provide the means to a reader to more easily see the interactions and effects of each treatment. Even though clear, the reader must spend relatively lot of time to compare and to follow the gained results.

Discussion:

Discussion should be put in sub-headings according to the main research questions/hypotheses to be more easily followed.

L 297-299 Statement is unnecessarily repeated from the Introduction part. Decide on place in the manuscript where it is important to be introduced to the reader.

In addition, the suggestion (not requisite) to the authors may also be to think about discussing about the dynamics of the seedling development. The speed seedling is growing (for different morphological parameter )could be indication of its vitality and interesting to mention here. Even not crucial, dynamic sometimes can reflect some important processes or influence of investigated treatments on seedlings.

References:

Decide if you are going to use space or not between the initials of authors – harmonize.

L 448 Delete space after “2015, 51 (6) “, and “:” into “,”

L 443 Change “Ostonen, I.,” with “Ostonen, I.;”

L 387, L477 Add space after “2010,”

L 434 Add space after “2011,”

Author Response

(The authors gave the same response as above.)

Reviewer 3 Report

Great work. A well written manuscript with well designed experiments and solid data. 

My questions and comments as follow and in the attached file: 

Line 2-3: Please find the editorial comment on the manuscript on the yellow highlighted letters

Line 43: Should be 'The high'

Line 103: Since the seed lot was collected in 2014, Please mention the storage condition of the seeds.

Line 104: How many seeds were planted to begin with?

Line 104: Please mention the percent moisture content of the sand bed.

Line 106: These 30% sprouted seeds were transplanted? or the rest of the non-germinated seeds also transplanted?

Line 107: how many seeds or sprouted seeds transplanted in each container or cell? 

Line 127: by the gravimetric water content  (GWC)

Line 129: The irrigation schedule was determined by GWC [40]. 

Line 141: The subsamples 

Line 141: were

Line 143: repeated text must be deleted 

Line 148: Specific root length (SRL)

Line 158: 'Tukey' not Turkey

Line 178: Please delete the 'are presented below'

Line 219: Figure 2B. (bold)

Line 268: Throughout the discussion please mention the species name that other researches worked on! Example: Doi et al work was on Chamaecyparis obtusa 

Line 326: 

Suggestions:

increase seedlings uniformity with similar height, root...

or 

...produce seedlings with higher uniformity in biometric parameters such as height, root diameter and root volume without effect on coarse roots...

Author Response

(The authors gave the same response as above.)
